# TEFormer: Structured Bidirectional Temporal Enhancement Modeling in Spiking Transformers

**Sicheng Shen** [1 2 3]  **Mingyang Lv** [1 4]  **Bing Han** [1]  **Dongcheng Zhao** [1]  **Guobin Shen** [1]  **Feifei Zhao** [1]  **Yi Zeng** [1 3]

## Abstract

In recent years, Spiking Neural Networks (SNNs) have achieved remarkable progress, with Spiking Transformers emerging as a promising architecture for energy-efficient sequence modeling. However, existing Spiking Transformers still lack a principled mechanism for effective temporal fusion, limiting their ability to fully exploit spatiotemporal dependencies. Inspired by feedforward–feedback modulation in the human visual pathway, we propose **TEFormer**, the first Spiking Transformer framework that achieves bidirectional temporal fusion by decoupling temporal modeling across its core components. Specifically, TEFormer employs a lightweight and hyperparameter-free **forward temporal fusion mechanism in the attention module**, enabling fully parallel computation, while incorporating a **backward gated recurrent structure in the MLP** to aggregate temporal information in reverse order and reinforce temporal consistency. Extensive experiments across a wide range of benchmarks demonstrate that TEFormer consistently and significantly outperforms strong SNN and Spiking Transformer baselines under diverse datasets. Moreover, through the first systematic evaluation of Spiking Transformers under different neural encoding schemes, we show that the performance gains of TEFormer remain stable across encoding choices, indicating that the improved temporal modeling directly translates into reliable accuracy improvements across varied spiking representations. These results collectively establish TEFormer as an effective and general framework for temporal modeling in Spiking Transformers. Code is available here.

[1]Institute of Autmation, CAS [2]School of Future Tech., UCAS [3]Zhongguancun Academy [4]School of AI., UCAS. Correspondence to: Feifei Zhao <zhaofeifei2014@ia.ac.cn>, Yi Zeng <yi.zeng@ia.ac.cn>.

*Proceedings of the 43rd International Conference on Machine Learning*, Seoul, South Korea. PMLR 306, 2026. Copyright 2026 by the author(s).

## 1. Introduction

Spiking Neural Networks (SNNs) are increasingly regarded as a promising foundation for next-generation artificial intelligence. By representing information with sparse binary spikes, SNNs inherently offer advantages in energy efficiency, computational throughput, and hardware compatibility (Maass, 1997; Zeng et al., 2023), making them particularly attractive for neuromorphic intelligence. This potential has been further amplified by recent advances in neuromorphic processors (Roy et al., 2019) and event-driven sensing technologies (Gallego et al., 2020), which provide both efficient computing substrates and natural input modalities for spiking computation. Together, these developments elevate SNNs from purely biologically inspired models to a practical and scalable paradigm for large-scale intelligent systems. Early SNN models were primarily driven by neuroscience insights and biologically faithful designs (Cheng et al., 2020; Zhao et al., 2020; Dong et al., 2023). While such approaches offer strong interpretability, they face substantial challenges in scaling to deep architectures and supporting efficient training. To overcome these limitations, subsequent research increasingly adopted architectural advances from ANNs, leading to spiking counterparts of mainstream models such as CNNs, RNNs, and GNNs (Fang et al., 2021a; Hu et al., 2021; Lotfi Rezaabad & Vishwanath, 2020; Zhu et al., 2022). These efforts significantly improved performance and scalability, accelerating the development of modern SNN architectures.

Despite these advances, most existing SNN architectures remain task-specific and lack a unified modeling paradigm. In contrast, Transformers exhibit strong scalability, effective sequence modeling capability, and natural extensibility to multimodal and large-scale settings (Vaswani et al., 2017; Dosovitskiy et al., 2020). Introducing this paradigm into the spiking domain is therefore a critical step toward general-purpose neuromorphic intelligence. This motivation has led to the emergence of Spiking Transformers, which aim to combine the sparse, event-driven computation of SNNs with the expressive modeling capacity of Transformers. Since the introduction of Spikformer (Zhou et al., 2022), a growing body of work has steadily enriched this research direction, including the Spike-driven Transformer family (Yao

et al., 2023; 2024), DISTA (Xu et al., 2023; 2024), QK-Former (Zhou et al., 2024a), and Spike2Former (Lei et al., 2025). More recently, STEP (Shen et al., 2025) has provided a unified evaluation framework, promoting reproducibility and fair comparison. However, while Spiking Transformers have shown promising performance, their ability to model temporal dependencies remains limited. Recent works, such as TIM (Shen et al., 2024), STAtten (Lee et al., 2025), and ST-SSA (Zhou et al., 2025), attempt to enhance temporal dynamics by introducing explicit temporal aggregation or recurrent operations within the attention mechanism. These designs either rely on heuristic, task-sensitive hyperparameters or impose sequential dependencies across time steps, thereby compromising the inherent parallelism of attention and increasing training and inference latency. In contrast, the human visual pathway exhibits pervasive feedforward–feedback modulation, where temporal information is integrated in a fundamentally **bidirectional manner** (Saban & Gabay, 2023; Scott, 2004). The lack of mechanisms that reflect this property suggests that temporal modeling in Spiking Transformers remains insufficiently explored, motivating more principled designs that can capture bidirectional temporal dependencies without sacrificing computational efficiency.

To address these challenges, we propose a **Temporal Enhanced Spiking Transformer (TEFormer)**, which adopts a structured and lightweight temporal modeling strategy. Instead of introducing sequential operations into attention, TEFormer integrates a hyperparameter-free module into spiking attention to enable **parallel forward temporal fusion**, preserving computational efficiency. In addition, we redesign the upsampling pathway of the multi-layer perceptron to support **backward temporal tracing**, complementing forward attention with reverse-time information aggregation. Together, these components form a unified framework for **bidirectional temporal fusion** in Spiking Transformers, advancing temporal modeling beyond prior unidirectional and heuristic approaches.

The main contributions of this work are summarized as follows:

- We propose a **Temporal Enhanced Attention (TEA)** module that enables parallel forward temporal fusion in spiking attention without additional hyperparameters, improving temporal modeling while preserving Transformer efficiency.

- We introduce a gated temporal mechanism in a restructured MLP (**T-MLP**) to support backward temporal tracing, which complements forward attention and enables **bio-inspired** effective bidirectional temporal fusion in Spiking Transformers.

- Extensive experiments across multiple datasets and

neuron encoding schemes demonstrate that the proposed bidirectional temporal fusion yields robust and transferable performance gains, validating the effectiveness of TEFormer.

## 2. Related Work

### 2.1. Temporal Enhancement in SNNs

Enhancing temporal dynamics is a central theme in improving information processing in spiking neural networks. At the neuron level, prior work has focused on strengthening temporal integration by introducing learnable dynamics. PLIF (Fang et al., 2021b) incorporates learnable membrane time constants to enhance temporal accumulation, while IIRSNN (Fang et al., 2020) improves spatiotemporal processing through explicit synaptic modeling. In addition, a series of LIF neuron variants with learnable dynamics further extend the capability of modeling long-term dependencies.

At the network level, TA-SNN (Yao et al., 2021) reinforces temporal information by adaptively weighting time steps, TCJA (Zhu et al., 2024) proposes joint attention for adaptive temporal modeling, and STSC-SNN (Yu et al., 2022) combines temporal convolution with attention mechanisms, offering network-level solutions to temporal enhancement.

Temporal enhancement in Spiking Transformers has been explored in only a few representative works. TIM (Shen et al., 2024) strengthens temporal information via step-wise convolution on queries, while STAtten (Lee et al., 2025) and ST-SSA (Zhou et al., 2025) design specialized attention mechanisms to improve temporal expressiveness. Despite their effectiveness, these methods share common characteristics: temporal information is mainly propagated in a forward direction, key design choices rely on heuristic and dataset-sensitive hyperparameters, and temporal modeling is tightly coupled with attention computation, which may limit parallel efficiency. These observations indicate that temporal modeling in Spiking Transformers remains an open and evolving research direction.

### 2.2. Frontiers in Spiking Transformers

Brain-inspired Spiking Neural Networks (SNNs) emulate spike-based information transmission and synaptic plasticity mechanisms of biological neural systems, enabling low power consumption and high computational efficiency. These characteristics make SNNs naturally well suited for neuromorphic hardware, while the introduction of surrogate gradient methods has made the training of large-scale SNN models feasible (Neftci et al., 2019). As the field matures, Spiking Transformers have emerged as a promising direction, integrating global modeling capability with energy-efficient spiking computation.

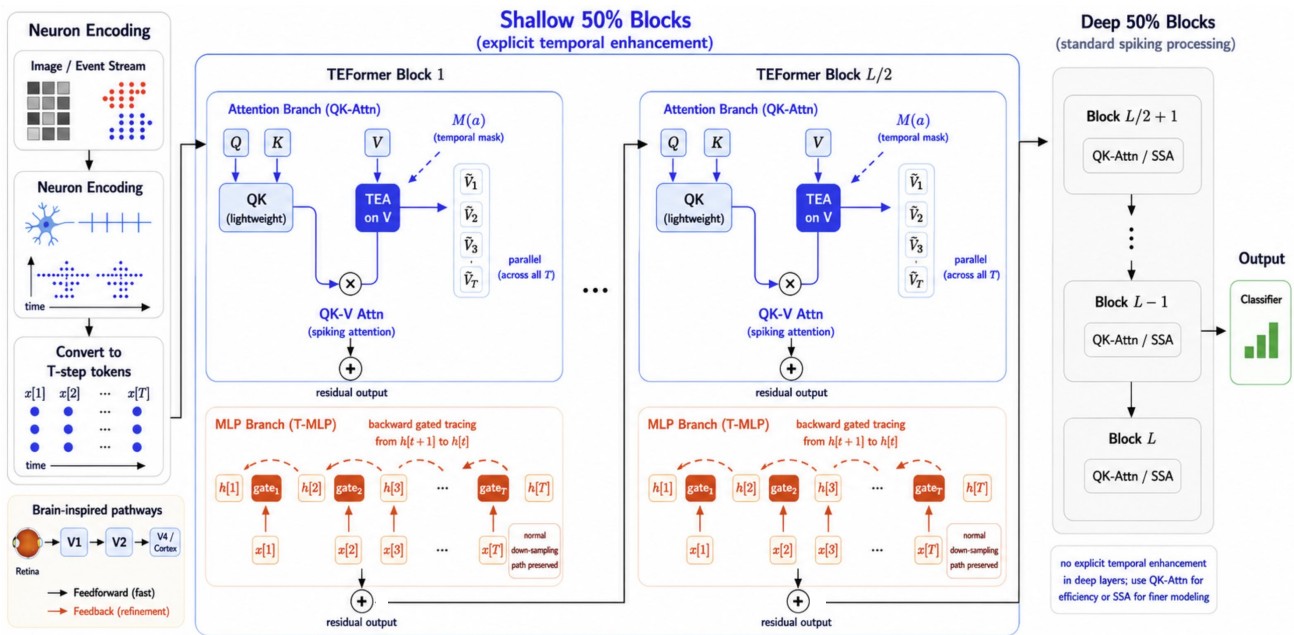

*Figure 1.* The pipeline of TEFormer with brain-inspired perspective.

Spikformer (Zhou et al., 2022) initiated this line of research by introducing spiking attention without Softmax. Subsequent works further advanced this paradigm: SDT (Yao et al., 2023) reduced attention complexity to linear time via sparse masking, QKFormer (Zhou et al., 2024a) improved accuracy through QK attention and hierarchical design, and SpikingResformer (Shi et al., 2024) enhanced efficiency with multi-stage ResNet structures and Dual-Spike attention. More recently, SNN-ViT (Wang et al., 2025) incorporated biologically inspired Saccadic Spike Self-Attention for dynamic visual selection.

Beyond classification, Spiking Transformers have been extended to a broader range of tasks. SDT-v2 (Yao et al., 2024) explored segmentation and detection, while Spike2Former (Lei et al., 2025), Spiking Point Transformer (Wu et al., 2025), and SpikingViT (Yu et al., 2024) addressed segmentation, 3D point cloud processing, and object detection, respectively. STEP (Shen et al., 2025) further provided a unified BrainCog-based framework, highlighting the scalability and general applicability of Spiking Transformers for neuromorphic intelligence.

## 3. Preliminary

### 3.1. Spiking Neurons

Spiking neural networks (SNNs) employ bio-inspired neurons that communicate via discrete spikes rather than continuous activations. In this work, we adopt the widely used **Leaky Integrate-and-Fire (LIF) neuron** (Hunsberger & Eliasmith, 2015), which integrates synaptic currents over

time with a leaky membrane potential:

$$\tau_m \frac{dV(t)}{dt} = -\big(V(t) - V_{\text{rest}}\big) + RI(t), \tag{1}$$

where $\tau_m$ is the membrane time constant, $V_{\text{rest}}$ the resting potential, and $I(t)$ the input current. A spike is generated when $V(t)$ exceeds a threshold $V_{\text{th}}$, producing a binary output $S(t) \in \{0, 1\}$ and resetting the membrane potential. This temporal accumulation and reset mechanism provides the foundation for capturing fine-grained dynamics in spiking architectures such as spiking transformers.

### 3.2. Neuron Encoding Schema

In SNNs, information is represented by discrete spike events and their temporal dynamics. However, most standard benchmarks, such as MNIST and CIFAR, are static, requiring neuron encoding to convert real-valued inputs into temporally structured spike representations.

Several encoding schemes have been proposed for static data, including rate, time-to-first-spike (TTFS), phase, and direct encoding (Adrian, 1926; Park et al., 2020; Kim et al., 2018). While direct encoding is widely adopted for its simplicity and efficiency, it largely suppresses temporal variation across time steps, limiting the exploitation of temporal dynamics. In contrast, more brain-inspired encoding schemes introduce step-wise temporal diversity. Consequently, the impact of encoding choices on temporal modeling and performance has not been systematically studied under a unified SNN framework. See details in Appendix. A.1.

### 3.3. Spiking Attention

Spiking Self-Attention (SSA) (Zhou et al., 2022) introduces the attention mechanism into Spiking Neural Networks by replacing softmax-based interactions with spiking-compatible operations. While SSA enables expressive token-level modeling, its computational cost grows quadratically with the number of tokens. To improve efficiency, QK-Attention (Zhou et al., 2024a) simplifies spiking attention by removing explicit token–token interactions, substantially reducing computational complexity.

In this work, we leverage the complementary strengths of these two formulations: QK-Attention is adopted in shallow layers to maintain high computational efficiency, while SSA is employed in deeper layers to facilitate more fine-grained spatiotemporal modeling. For a detailed formulation and comparison of these spiking attention variants, please refer to Appendix A.2.

## 4. Methodology

### 4.1. TEFormer

**Inspired by the feedforward and feedback pathways in the human visual system**, we propose TEFormer (Fig. 1 (a)). Temporal fusion is primarily applied in the shallow stages of the network, where representations emphasize low-level spatiotemporal patterns. Accordingly, TEFormer introduces explicit temporal modeling in the shallow layers (the first 50% of Transformer layers), where bidirectional temporal fusion is constructed via paired cascades of a forward Temporal Enhancement Ahead (TEA, Sec. 4.2) module and a backward Temporal MLP (T-MLP, Sec. 4.3).

Moreover, neurophysiological evidence suggests that feedforward processing in the visual pathway is generally faster than feedback processing (Aggarwal et al., 2022). Motivated by this observation, we place particular emphasis on designing a synchronous and efficient forward temporal enhancement mechanism via TEA, while T-MLP is responsible for more refined temporal modeling along the backward pathway. This asymmetric yet coordinated design enables effective bidirectional temporal interaction while maintaining computational efficiency.

### 4.2. Temporal Enhancement Attention

TEA is designed as a lightweight and fully parallel temporal fusion module for Spiking Transformers. Unlike existing temporal enhancement methods that introduce multiple dataset-sensitive hyperparameters or require step-wise sequential processing, TEA is parameterized by only a *single learnable scalar* and does not involve any manually defined

hyperparameters (Fig. 1 (b)).

$$\mathbf{V} \in \mathbb{R}^{TB \times C \times H \times W},$$
$$\alpha = 0.5 + 0.5 \cdot \sigma(\theta), \quad \theta \in \mathbb{R},$$
$$M_{i,j} = \alpha(1-\alpha)^{i-j}\,\mathbb{I}(i \geq j) + (1-\alpha)^i\,\mathbb{I}(j=0), \quad (2)$$
$$\widetilde{\mathbf{V}} = \mathbf{M}\mathbf{V}_t,$$
$$\widetilde{\mathbf{V}} \in \mathbb{R}^{TB \times C \times H \times W}.$$

Specifically, TEA performs forward temporal fusion by computing a set of weights over all historical time steps using an exponential moving average controlled by a single scalar parameter $\theta$. The corresponding temporal decay factor $\alpha$ is obtained via a sigmoid transformation, ensuring numerical stability and boundedness. As shown in Eq. 2, the temporal weights are pre-constructed into a temporal mask matrix, which is applied to the input sequence through a single matrix multiplication. This formulation enables temporal fusion across all time steps in a synchronized and fully parallel manner, without introducing any recurrent or step-wise dependencies. Importantly, TEA is agnostic to the specific attention branch and can be flexibly applied to any of the $Q$, $K$, or $V$ pathways. In this work, TEA is instantiated as the temporal branch on the value pathway and integrated with QK Attention, allowing temporal enhancement to be achieved without modifying the core attention computation. By avoiding additional convolutional or linear layers and leveraging the linear-complexity structure of QK Attention, TEA introduces negligible computational overhead while preserving efficient and scalable attention processing.

Overall, TEA provides a minimal yet effective temporal enhancement mechanism that aligns well with the design principles of Transformer architectures, offering improved temporal modeling capacity with minimal parameter and computational cost.

### 4.3. Temporal MLP for Backward Fusion

Most existing temporal enhancement methods in SNNs and Spiking Transformers adopt unidirectional temporal fusion, where information propagates only from past to future. While effective for capturing historical dependencies, such designs neglect future-to-present constraints that are often crucial for globally consistent temporal representations. To address this limitation, we propose a Temporal MLP (T-MLP) with a backtracking mechanism, enabling efficient bidirectional temporal fusion. Instead of conventional temporal up-sampling, our design employs a gated recurrent update that allows future information to flow backward with minimal computational overhead (Fig. 1 (c)).

Given an input sequence $\{\mathbf{X}_t\}_{t=0}^{T-1}$, the hidden state is initialized from the last time step as

$$\mathbf{h}_{T-1} = \text{LIF}(\mathbf{W}_{\text{in}}\mathbf{X}_{T-1}). \quad (3)$$

For each preceding time index $r = T - 2, \ldots, 0$, the hidden state is updated through a gated backward recurrence:

$$
\begin{aligned}
\mathbf{h}_r = \mathrm{LIF}\Big( & \sigma\big(\mathbf{W}_{fx}\mathbf{X}_r + \mathbf{W}_{fh}\mathbf{h}_{r+1}\big) \odot \mathbf{h}_{r+1} \\
& + \big(1 - \sigma(\cdot)\big) \odot \mathbf{W}_{\mathrm{in}}\mathbf{X}_r\Big),
\end{aligned}
\tag{4}
$$

where $\sigma(\cdot)$ denotes a sigmoid gate that adaptively balances future hidden states and current inputs. To preserve the computational efficiency of SNNs, we adopt a single-gate design, avoiding the overhead of multi-branch gating mechanisms.

After obtaining the backward-enhanced hidden sequence $\{\mathbf{h}_t\}_{t=0}^{T-1}$, the output at each time step is computed as

$$
\mathbf{Y}_t = \mathrm{LIF}\Big(\mathrm{BN}(\mathbf{W}_o\mathbf{h}_t)\Big), \quad t = 0, \ldots, T-1. \tag{5}
$$

Overall, the proposed T-MLP replaces the conventional up-sampling module with a gated backtracking update while preserving the down-sampling structure. By explicitly introducing future-to-past interactions with minimal gating complexity, it achieves bidirectional temporal fusion and strengthens temporal representation in spiking transformers without sacrificing efficiency.

## 5. Experiment

All experiments are conducted on the unified evaluation platform provided by STEP (Shen et al., 2025), which is built upon the BrainCog framework (Zeng et al., 2023). This platform offers a consistent training and evaluation pipeline for Spiking Transformers, ensuring fair comparisons across static datasets, neuromorphic datasets, and datasets with complex temporal structures by sharing identical data loading, augmentation, and training configurations.

### 5.1. Static Dataset

Static datasets constitute a fundamental benchmark for evaluating vision models. In Spiking Neural Networks, static inputs are typically processed using direct encoding, which provides limited explicit temporal variation. As a result, introducing temporal enhancement into SNNs under static settings is non-trivial and often leads to performance degradation. Notably, existing temporal enhancement methods for Spiking Transformers, such as TIM and ST-SSA, do not report results on static datasets, highlighting the challenge of effectively leveraging temporal modeling in this scenario.

We evaluate TEFormer on three widely used static benchmarks: CIFAR10, CIFAR100 (Krizhevsky et al., 2009), and SVHN (Netzer et al., 2011) (Appendix B.1). All models are trained under the same experimental setup for 400 epochs on a single NVIDIA Tesla A100 GPU. For a fair comparison, all architectures are configured with the same model size (4 Transformer blocks with an embedding dimension

of 384). For TEFormer, bidirectional temporal fusion (TEA + T-MLP) is applied only in the first two blocks, while the remaining blocks adopt conventional modules.

As shown in Tab. 1, TEFormer consistently outperforms all baseline methods and is the only model achieving accuracy above 96% on CIFAR10. To enable a direct comparison, we re-implement TIM using Spikformer as the backbone. Despite the increased model capacity, TIM exhibits a pronounced performance drop on static datasets, indicating that its step-wise, unidirectional temporal enhancement does not generalize well to static inputs. In contrast, the strong and stable performance of TEFormer demonstrates that bidirectional temporal fusion can effectively enhance representation learning without introducing detrimental temporal bias.

To further verify that the observed improvements are not specific to a particular model configuration, we conduct additional experiments across multiple model scales. As summarized in Tab. 12, TEFormer consistently achieves superior performance under varying model sizes. These results confirm that the proposed bidirectional temporal enhancement remains effective even in conventional static settings, underscoring the importance of balanced forward–backward temporal modeling.

### 5.2. Neuromorphic Datasets

Neuromorphic datasets collected by dynamic vision sensors (DVS) provide a natural and challenging testbed for spiking neural networks, as they consist of sparse, asynchronous event streams with fine-grained temporal resolution.

Leveraging the STEP framework, we evaluate TEFormer on three representative neuromorphic benchmarks: CIFAR10-DVS, N-CALTECH101, and NCARS (Li et al., 2017; Orchard et al., 2015; Sironi et al., 2018). As summarized in Tab. 2, TEFormer consistently outperforms prior spiking Transformer baselines across datasets and model scales. On CIFAR10-DVS, it achieves 81.90%, showing a clear improvement over existing methods. On N-CALTECH101, TEFormer obtains the best performance under the standard configuration and maintains its advantage when scaled to larger model sizes, such as 4-384. On the high-accuracy NCARS benchmark, it reaches 95.95%, matching the strongest baseline. These results demonstrate TEFormer's effectiveness and scalability for neuromorphic vision tasks. These results demonstrate that the proposed bidirectional temporal fusion enables more effective utilization of event-based temporal cues, leading to stable and scalable improvements on diverse neuromorphic benchmarks. This highlights the suitability of TEFormer for event-driven recognition tasks and reinforces the importance of explicit temporal modeling in spiking Transformers.

*Table 1.* Performance (Acc@1) of ANN and SNN models, particularly Spiking Transformers, on the CIFAR10/100 datasets. ∗ denotes models reproduced by the authors based on the STEP framework; † denotes experimental results obtained in this paper through unified evaluation under the STEP framework.

| Model | Architecture | Step | SNN | CIFAR10 | CIFAR100 |
|---|---|---|---|---|---|
| ResNet-19 (He et al., 2016) | ResNet | 1 | ✘ | 94.97 | 75.35 |
| PLIF (Fang et al., 2021b) | ConvNet | 8 | ✘ | 93.5 | 74.8 |
| tdBN (Zheng et al., 2021) | ResNet | 4 | ✓ | 92.92 | - |
| DIET-SNN (Rathi & Roy, 2020) | VGGNet | 5 | ✓ | 92.70 | 69.67 |
| DSR (Meng et al., 2022) | ResNet | 20 | ✓ | 95.40 | 78.50 |
| SNASNet (Kim et al., 2022) | ConvNet | 5 | ✓ | 93.64 | 73.04 |
| ViT (Dosovitskiy et al., 2020)[†] | ViT | 1 | ✘ | 90.89 | - |
| Spikingformer (Zhou et al., 2026)[†] | Spikingformer | 4 | ✓ | 95.53 | 79.12 |
| Spikformer+SEMM (Zhou et al., 2024b)[†] | Spikformer | 4 | ✓ | 94.98 | 77.59 |
| Spiking Wavelet (Fang et al., 2024)[†] | SWFormer | 4 | ✓ | 95.31 | 76.99 |
| Spikformer (Zhou et al., 2022)[∗] | Spikformer | 4 | ✓ | 95.09 | 77.72 |
| SDT (Yao et al., 2023)[∗] | SD-Transformer | 4 | ✓ | 95.78 | 78.64 |
| QKFormer (Zhou et al., 2024a)[∗] | H-Spikformer | 4 | ✓ | 95.91 | 79.09 |
| TIM(Shen et al., 2024)[∗] | Spikformer | 4 | ✓ | 94.20 | 75.04 |
| **TEFormer(ours)** | H-Spikformer | 4 | ✓ | **96.24** | **79.84** |

## 5.3. Temporal Complicated Datasets

Temporally complicated datasets impose stricter requirements on temporal modeling, as they involve long-range dependencies, complex dynamics, or both. sCIFAR and sMNIST are constructed by serializing static images into long sequences (Chang et al., 2017), explicitly challenging a model's ability to integrate information over extended temporal horizons. In contrast, SHD, HMDB51-DVS, and UCF101-DVS correspond to speech recognition, event-based action recognition, and video classification tasks, respectively, featuring richer and more realistic temporal dynamics (Bi et al., 2020; Cramer et al., 2020). Compared with standard neuromorphic or static classification benchmarks, these datasets present substantially higher temporal complexity.

As shown in Tab. 3, TEFormer consistently achieves the best performance across all evaluated datasets. On serialized benchmarks, it improves upon strong Transformer-based baselines on both sCIFAR and sMNIST, indicating more effective long-range temporal integration. On SHD, TEFormer yields a clear margin over prior spiking models, demonstrating its advantage in speech-oriented temporal modeling. For event-based action recognition tasks, it achieves competitive or superior accuracy on HMDB51-DVS and UCF101-DVS, outperforming existing spiking Transformers and conventional video baselines.

Overall, these results show that the proposed bidirectional temporal fusion enables robust and consistent improvements across diverse temporal regimes, ranging from synthetic long-sequence inputs to real-world speech and action recognition. This confirms the effectiveness and generality of TEFormer in handling complex temporal dependencies.

## 6. Discussion

### 6.1. Spiking Transformer Among Various Encoding Schema

Traditional SNNs commonly process static datasets using **direct encoding** (Eq. 6), which has become the dominant encoding scheme in current Spiking Transformer studies due to its simplicity and effectiveness. It repeats the same static input for $T$ timesteps, allowing temporal accumulation but also introducing **severe temporal redundancy**, since different timesteps contain nearly identical signals. As a result, direct encoding provides limited temporal diversity and is insufficient for comprehensively evaluating temporal modeling capability.

In contrast, phase, rate, and time-to-first-spike (TTFS) encodings transform static inputs into dynamic spike sequences. By constructing **explicit differences across timesteps**, these schemes introduce richer temporal variations and make static recognition a more challenging **se-**

*Table 2.* Performance (Acc@1) of different models on NCAL, NCARS, and CIFAR10-DVS datasets. All experiments were conducted **without** NDA.

| Dataset | Model | Size | Acc@1 | Step | Batch-Size |
|---------|-------|------|-------|------|------------|
| **CIFAR10-DVS** | Spikformer (Zhou et al., 2022) | 2-256 | 80.25 | 10 | 32 |
| | QKFormer (Zhou et al., 2024a) | 2-256 | 79.30 | 10 | 32 |
| | TIM (Shen et al., 2024) | 2-256 | 80.85 | 10 | 32 |
| | **TEFormer(ours)** | 2-256 | **81.90** | 10 | 32 |
| **N-CALTECH101** | Spikformer | 2-256 | 77.93 | 10 | 16 |
| | QKFormer | 2-256 | 76.09 | 10 | 16 |
| | **TEFormer(ours)** | 2-256 | **78.50** | 10 | 16 |
| | Spikformer | 4-384 | 73.33 | 16 | 16 |
| | QKFormer | 4-384 | 76.09 | 16 | 16 |
| | **TEFormer(ours)** | 4-384 | **78.05** | 16 | 16 |
| **NCARS** | Spikformer | 2-256 | 95.60 | 16 | 32 |
| | QKFormer | 2-256 | 95.29 | 16 | 32 |
| | **TEFormer(ours)** | 2-256 | **95.95** | 16 | 32 |

*Table 3.* Performance (Acc@1) on Temporal Complicated datasets.

| Model | sCIFAR | sMNIST | SHD | HMDB51-DVS | UCF101-DVS |
|-------|--------|--------|-----|------------|------------|
| C3D (Tran et al., 2015) | - | - | - | 41.7 | 47.2 |
| P3D-63 (Qiu et al., 2017) | - | - | - | 40.4 | 53.4 |
| QKFormer (Zhou et al., 2024a) | 80.20 | 94.25 | 88.56 | 63.53 | 61.31 |
| TIM (Shen et al., 2024) | - | - | 85.24 | 59.68 | 61.18 |
| **TEFormer(ours)** | **80.94** | **96.20** | **90.19** | **63.65** | **63.16** |

**quence modeling problem**. Under such settings, models must integrate temporally distributed information rather than relying mainly on repeated spatial inputs.

Despite the importance of encoding strategies, existing Spiking Transformers have not been systematically evaluated under different encoding schemes. To the best of our knowledge, TEFormer is the first work to conduct such a comprehensive evaluation. As shown in Tab. 4, TEFormer consistently outperforms all baselines under direct, phase, rate, and TTFS encodings, with especially clear gains under dynamic encodings. These results indicate that TEFormer's improvements come from its **enhanced temporal modeling capability**, and further suggest that **encoding-based evaluation** can serve as a useful paradigm for assessing temporal modeling in SNNs.

### 6.2. Ablation Study

#### 6.2.1. BI-DIRECTIONAL TEMPORAL FUSION NECESSITY

**Module Necessity** In TEFormer, we design two key modules: (i) Temporal MLP (T-MLP) and (ii) Temporal Enhance

*Table 4.* Acc@1 (%) on CIFAR10 under different neuron encodings (Step=4).

| Model | Direct | Phase | Rate | TTFS |
|-------|--------|-------|------|------|
| Spikformer (Zhou et al., 2022) | 95.09 | 82.63 | 82.68 | 81.87 |
| SDT (Yao et al., 2023) | 95.78 | 85.33 | 84.06 | 84.52 |
| QKFormer (Zhou et al., 2024a) | 95.91 | 87.76 | 83.77 | 84.69 |
| TIM (Shen et al., 2024) | 94.20 | 81.43 | 81.48 | 80.66 |
| **TEFormer** | **96.24** | **89.92** | **84.74** | **87.46** |

Attention (TEA), which are integrated into the MLP and Attention components, respectively. These two modules fuse temporal information from opposite directions and complement each other to jointly enhance the model's temporal modeling capability. We first aim to demonstrate that the introduction of these modules is necessary, and that their coordinated collaboration yields a synergistic effect, achieving performance gains beyond a simple additive combination. As shown in Table 5, introducing T-MLP alone yields only marginal gains, while a single temporal module may even degrade performance. In contrast, jointly applying both tem-

*Table 5.* Ablation study for module necessity under CIFAR10 with 4-384 size trained with 400 epochs.

| Model | Size | Acc@1 (%) | Steps |
|---|---|---|---|
| Baseline | 4-384 | 95.91 | 4 |
| Baseline + T-MLP | 4-384 | 95.98 | 4 |
| Baseline + TEA | 4-384 | 95.85 | 4 |
| **TEFormer** | 4-384 | **96.24** | 4 |

poral components leads to a clear improvement, indicating that effective temporal modeling in TEFormer arises from their coordinated interaction rather than isolated enhancements.

**Bi-directional Fusion Analysis** TEFormer achieves bidirectional temporal enhancement by pairing TEA with T-MLP, where TEA performs forward temporal fusion while T-MLP is responsible for reverse temporal integration. Such a directional design is not arbitrary. To justify the necessity of this architectural choice, we conduct an ablation study to examine whether assigning these two modules to opposite temporal directions is indeed reasonable, and whether bidirectional temporal enhancement is essential for effective temporal modeling. Through controlled comparisons, we aim to demonstrate that neither direction alone is sufficient, and that meaningful performance gains emerge only when forward and reverse temporal fusion are jointly considered. Based on the results in Tab. 6, where the direction of arrow

*Table 6.* Ablation study for bi-directional fusion under CIFAR10 with 4-384 size trained with 400 epochs.

| TEA | T-MLP | Acc@1 (%) | Steps |
|---|---|---|---|
| $\leftarrow$ | $\rightarrow$ | 96.06 | 4 |
| $\leftarrow$ | $\leftarrow$ | 96.09 | 4 |
| $\rightarrow$ | $\rightarrow$ | 95.95 | 4 |
| $\rightarrow$ | $\leftarrow$ | **96.24** | 4 |

denotes fusion direction, we can conclude that the fusion directions assigned to the two modules in TEFormer constitute the optimal design. Directionally consistent temporal fusion consistently outperforms its unidirectional counterparts, demonstrating that bidirectional temporal fusion is both reasonable and effective. Under identical parameter budgets, TEFormer maximizes temporal gains by jointly leveraging forward and reverse temporal integration.

## 6.3. Gate Selection

To further investigate the design choice of the temporal gate in T-MLP, we compare the proposed single-gate mechanism with two representative alternatives: a GRU-style gated update and an EMA-style temporal update. The GRU vari-

*Table 7.* Ablation study on different gate designs in T-MLP. All models are evaluated under the same training setting.

| Model | CIFAR10 | CIFAR100 | SVHN | Params |
|---|---|---|---|---|
| TEFormer-EMA | 96.05 | 79.55 | 96.72 | 6.85M |
| TEFormer-GRU | 95.90 | 79.15 | 96.79 | 9.43M |
| **TEFormer** | **96.24** | **79.84** | **96.88** | 7.77M |

ant introduces stronger recurrent modeling capacity with multiple gates, while the EMA variant adopts a simpler temporal smoothing mechanism. As shown in Table 7, the proposed single-gate design achieves the best performance on CIFAR10, CIFAR100, and SVHN, while using fewer parameters than the GRU-style variant. Although the EMA-style variant has the smallest parameter count, its accuracy is consistently lower than that of TEFormer. These results indicate that the proposed gate provides a better balance between temporal modeling capability, parameter efficiency, and computational simplicity, validating the effectiveness of using a lightweight single-gate mechanism for backward temporal fusion.

### 6.3.1. SINGLE HYPERPARAMETER

In TEA, we introduce a single learnable parameter $\alpha$, playing a role analogous to the $\alpha$ in TIM (Shen et al., 2024). Although prior work on TIM has demonstrated that $\alpha$ is dataset-sensitive, it is still necessary to justify whether fixing this parameter is sufficient or if making it learnable provides tangible benefits. As shown in Tab. 8, models with

*Table 8.* Analisys for Alpha. The content in parentheses represents the optimal alpha values ultimately learned by the model in the two preceding stages.

| Dataset | alpha | Acc@1 | Step | Batch-Size |
|---|---|---|---|---|
| **CIFAR10** | 0.5 | 96.16 | 10 | 128 |
| | 0.75 | 96.04 | 10 | 128 |
| | **ours (0.66/0.59)** | 96.24 | 10 | 128 |
| **CIFAR10-DVS** | 0.5 | 81.9 | 10 | 32 |
| | 0.75 | 81.2 | 10 | 32 |
| | **ours (0.5/0.5)** | 81.9 | 10 | 32 |

a learnable $\alpha$ consistently outperform those using fixed values across both conventional and neuromorphic datasets, clearly validating the necessity of learning this parameter. Moreover, we observe a clear relationship between the temporal resolution and the learned $\alpha$: when the number of steps is larger, the model tends to learn a relatively smaller $\alpha$, indicating a preference for retaining more historical information; conversely, with fewer steps, a larger $\alpha$ is learned, emphasizing features from the current time step. This adaptive behavior further supports the design choice of modeling $\alpha$ as a learnable parameter.

# 7. Efficiency, Design Rationale & Limitations

## 7.1. T-MLP Energy Consumption

**Efficiency analysis of T-MLP.** Although T-MLP introduces a gated backward temporal update, it replaces the standard MLP up-projection rather than adding an independent temporal module. To examine the efficiency–accuracy trade-off, we report MLP FLOPs, parameters, and estimated energy consumption on CIFAR10 in Table 9. The default TEFormer with MLP ratio 4 achieves the best accuracy, while introducing additional computation and energy cost. When reducing the MLP ratio to 2, TEFormer obtains FLOPs comparable to standard Spiking Transformer baselines, consumes less energy than Spikformer, QKFormer, and TIM, and still achieves higher accuracy. These results show that the gain of TEFormer does not simply come from a larger computational budget, and the proposed T-MLP can provide a favorable efficiency–accuracy trade-off under compute- or energy-constrained settings. Meanwhile, this indicates that TEFormer can still achieve competitive performance in certain energy- or resource-constrained scenarios.

*Table 9.* Efficiency analysis of T-MLP on CIFAR10. $E_{\mathrm{MAC}}$, $E_{\mathrm{AC}}$, and $E_{\mathrm{total}}$ denote the estimated energy of multiply-and-accumulate operations, accumulation operations, and their sum, respectively. MR refers T-MLP mlp ratios in TEFormer.

| Model | Acc. | MLP FLOPs | Params | $E_{\mathrm{MAC}}$ | $E_{\mathrm{AC}}$ | $E_{\mathrm{total}}$ |
|---|---|---|---|---|---|---|
| Spikformer | 95.09 | 0.604G | 9.32M | 2.78 mJ | 0.54 mJ | 3.32 mJ |
| SDT | 95.78 | 0.604G | 9.32M | 0.00 mJ | 1.09 mJ | **1.09 mJ** |
| TIM | 94.20 | 0.604G | 9.41M | 2.78 mJ | 0.54 mJ | 3.32 mJ |
| MR= 4 | **96.24** | 1.736G | 7.77M | 3.82 mJ | 1.36 mJ | 5.18 mJ |
| MR= 2 | 95.98 | 0.642G | **6.93M** | 1.91 mJ | **0.48 mJ** | 2.39 mJ |

## 7.2. Design Rationale

In TEFormer, temporal enhancement is applied only to the first half of Transformer layers for both representation and efficiency reasons. Shallow layers mainly capture low-level spatiotemporal patterns, where temporal fusion is most beneficial, while deeper layers focus more on high-level semantic refinement. Applying temporal mixing to all layers may therefore introduce redundancy and disturb fine-grained representations.

We further validate this choice by varying the number of temporally enhanced layers under the same model scale. As shown in Table 10, increasing enhanced layers does not yield monotonic gains. The two-layer setting, corresponding to the first 50% layers in the 4-block architecture, provides a strong accuracy–efficiency trade-off, while using more enhanced layers introduces extra parameters with only marginal or negative improvements.

*Table 10.* Ablation study on the number of layers equipped with temporal enhancement. All experiments are conducted with the same 4-block architecture and embedding dimension of 384.

| TE Layers | Model Size | CIFAR10 | CIFAR100 | Params |
|---|---|---|---|---|
| 1 | 4-384 | 95.24 | 79.00 | 6.96M |
| 2 | 4-384 | **96.04** | 79.84 | 7.77M |
| 3 | 4-384 | 95.82 | **79.96** | 11.01M |
| 4 | 4-384 | 95.66 | 78.11 | 14.25M |

## 7.3. Limitations

While TEFormer demonstrates strong temporal modeling capability and bio-interpretability, several limitations remain: i) **Evaluation scope**: Results are based on software simulation and classification tasks only; hardware deployment and extension to more complex tasks remain unexplored. ii) **Neuron encoding**: Although influential for temporal modeling, a unified encoding standard is still lacking in the spiking neural network community. iii) **Causal inference**: Since T-MLP uses backward temporal fusion, TEFormer targets window-based recognition rather than strictly causal online inference.

# 8. Conclusion

In this work, we present **TEFormer**, a Temporal-Enhanced Spiking Transformer that addresses a long-standing limitation in temporal modeling for spiking Transformer architectures. By jointly redesigning the two core components of Transformers—attention and MLP—we introduce, for the first time, a unified framework for **bidirectional temporal fusion** in Spiking Transformers that is both efficient and biologically grounded. Specifically, the proposed Temporal Enhanced Attention (TEA) enables forward temporal fusion with a single learnable parameter while fully preserving parallel computation, and the Temporal MLP (T-MLP) incorporates a lightweight gated backward recurrence to impose future-to-past temporal constraints, echoing feedback modulation observed in biological neural systems.

Extensive experiments on static datasets, neuromorphic benchmarks, and temporally complex tasks demonstrate that TEFormer consistently outperforms strong spiking Transformer baselines under identical training settings. Notably, the performance gains are robust across both event-driven and conventional static datasets, underscoring the generality of **biologically inspired bidirectional** temporal modeling. Furthermore, our systematic evaluation under multiple neuron encoding schemes reveals that effective temporal modeling is critical beyond direct encoding scenarios.

Overall, TEFormer provides a principled, efficient, and scalable approach to temporal enhancement in Spiking Transformers, offering a biologically plausible foundation for future research on brain-inspired temporal representation learning and general-purpose neuromorphic intelligence.

## Acknowledgment

This work is supported by the Strategic Priority Research Program of the Chinese Academy of Sciences (Grant No. XDB1010301), the National Natural Science Foundation of China (Grant No. 62576341 & No. 62406325). We thank Tianming Yang from the Institute of Neuroscience, CAS, for insightful discussions related to this work. We also sincerely thank Sicheng's West Highland White Terrier, *Theta*, for providing adorable photos used in the visualizations of this paper.

## Impact Statement

Our method proposes the first brain-inspired, bidirectionally temporally enhanced Spiking Transformer. In addition, it provides a new solution and perspective for the development of brain-inspired artificial intelligence and spiking neural networks. All experiments are conducted on publicly available datasets, and we do not anticipate any negative societal or ethical impacts arising from this work.

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

# A. Spiking Transformer

## A.1. Spiking Encoding Schema

**Direct Encoding**

$$S_t(\mathbf{p}) = x(\mathbf{p}), \qquad t = 1, \ldots, T, \tag{6}$$

where $x(\mathbf{p}) \in [0, 1]$ is the normalized pixel intensity at position $\mathbf{p}$.

**Phase Encoding**

$$S_t(\mathbf{p}) = \begin{cases} 2^{-(b+1)}, & \text{if } v_{7-b}(\mathbf{p}) = 1, \ b \equiv (t-1) \pmod 8, \\ 0, & \text{otherwise,} \end{cases} \tag{7}$$

where $v(\mathbf{p}) = \lfloor 256\, x(\mathbf{p}) \rfloor$ and $v_k$ is the $k$-th bit.

**Rate Encoding**

$$S_t(\mathbf{p}) \sim \text{Bernoulli}\big(x(\mathbf{p})\big), \qquad \mathbb{E}\big[S_t(\mathbf{p})\big] = x(\mathbf{p}). \tag{8}$$

**Time-to-First-Spike (TTFS) Encoding**

$$t^\star(\mathbf{p}) = 1 + \left\lfloor (1 - x(\mathbf{p}))\, T \right\rfloor, \tag{9}$$

$$S_t(\mathbf{p}) = \begin{cases} \dfrac{1}{t^\star(\mathbf{p})}, & t = t^\star(\mathbf{p}), \\ 0, & \text{otherwise.} \end{cases} \tag{10}$$

Each neuron fires once, with higher intensities producing earlier spikes.

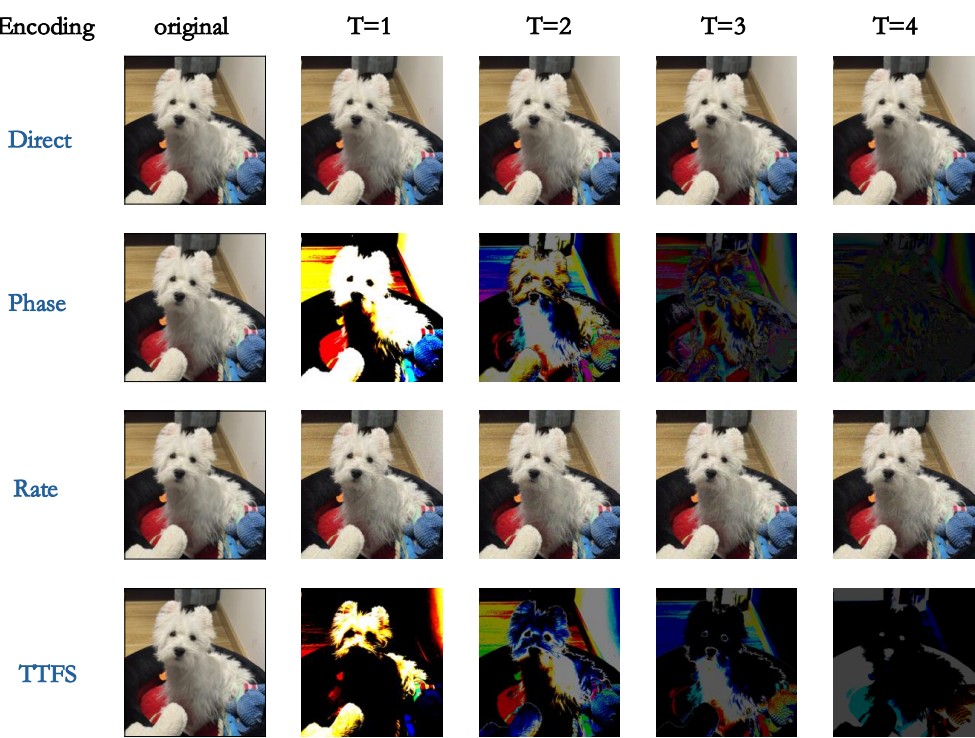

*Figure 2.* Caption

## A.2. Spiking Attention

This appendix provides a concise description of four representative spiking attention mechanisms, namely SSA, SDSA, ST-SSA, and TIM. For consistency, we denote the query, key, and value at time step $t$ as $Q[t]$, $K[t]$, and $V[t]$, respectively. The operator $\text{SN}(\cdot)$ represents a spiking neuron function (e.g., LIF), which introduces temporal dynamics and binary spike-based outputs.

**Spiking Self-Attention (SSA).** SSA is the most direct extension of conventional scaled dot-product attention into the spiking domain. The attention activation is computed as

$$A(\cdot) = QK^\top V \cdot \text{scale},$$

followed by a spiking neuron transformation

$$\text{SSA}(\cdot) = \text{SN}(A).$$

Compared with standard attention, SSA replaces the softmax normalization with spiking dynamics, enabling event-driven computation while preserving the core query–key–value interaction.

**Spike-Driven Self-Attention (SDSA).** SDSA modifies the attention computation to better align with spike-based processing by aggregating query–key interactions before combining with values. Specifically,

$$A(\cdot) = \text{SN}\big(\text{SUM}(QK^\top)\big), \qquad \text{SDSA}(\cdot) = VA \cdot \text{scale}.$$

This formulation reduces the computational burden of full matrix multiplications and emphasizes spike accumulation, making SDSA more hardware-friendly for neuromorphic implementations.

**Spatio-Temporal Spiking Self-Attention (ST-SSA).** ST-SSA explicitly models temporal correlations by introducing a recurrent memory term. The query is first transformed via a membrane-potential-based module,

$$M = \text{SN}(\text{MP}(Q)),$$

and combined with a temporal register $R$ to form

$$R_M[t] = M[t] \odot R, \qquad R_M = \text{AGG}(R_M[0], \ldots, R_M[t]).$$

The final attention output is computed as

$$\text{ST-SSA} = \text{SSA}(QR_M^\top, R_M K^\top, V) \cdot S,$$

where the scaling factor $S = s_1(1 - \sigma(\alpha)) + s_2\sigma(\beta)$ adaptively balances spatial and temporal contributions. This design allows ST-SSA to capture long-range temporal dependencies in spiking sequences.

**QK Attention** QKTA is an efficient spiking attention formulation that assigns token-wise importance scores without explicitly computing pairwise token interactions. By deriving a token attention vector from the query representation and using it to gate key features, QKTA substantially reduces computational complexity while preserving the core spiking attention behavior.

Given the token-wise input $\mathbf{X} \in \mathbb{R}^{N \times D}$, the spiking query and key representations are first computed as

$$\mathbf{Q} = \text{SN}(\mathbf{X}\mathbf{W}_Q), \qquad \mathbf{K} = \text{SN}(\mathbf{X}\mathbf{W}_K),$$

where $\text{SN}(\cdot)$ denotes the spiking neuron activation.

A token attention vector is obtained by aggregating the query features along the channel dimension:

$$\mathbf{s} = \mathbf{Q}\mathbf{1}_D \in \mathbb{R}^{N \times 1}, \qquad \mathbf{a} = \text{SN}(\mathbf{s}) \in \mathbb{R}^{N \times 1},$$

where $\mathbf{1}_D$ is an all-ones vector.

The token attention vector is converted into a binary token mask:

$$\mathbf{m} = g(\mathbf{a}) \in \{0, 1\}^{N \times 1},$$

where $g(\cdot)$ denotes a token selection function.

Finally, the output of QKTA is obtained by token-wise gating of the key representation:

$$\mathbf{X}' = \mathbf{m} \odot \mathbf{K} \in \mathbb{R}^{N \times D},$$

where $\odot$ denotes element-wise multiplication with broadcasting along the channel dimension.

**Temporal Interaction Module (TIM).** TIM focuses on temporal smoothing and integration of the query representation before applying spiking attention. A temporal feature extractor is defined as

$$f(x) = \text{SN}(\text{Conv}(x)),$$

and the query is updated recursively:

$$Q^{\text{TIM}}[t] = (1 - \alpha)Q[t] + \alpha f(Q^{\text{TIM}}[t - 1]), \qquad Q[t] = Q^{\text{TIM}}[t].$$

The final attention output is then obtained via

$$\text{TIM}(\cdot) = \text{SSA}(Q, K, V).$$

By introducing temporal inertia through recursive filtering, TIM enhances robustness to noise and improves temporal consistency in spiking attention outputs.

# B. Experiment

## B.1. SVHN Dataset

The Street View House Numbers (SVHN) dataset is a large-scale real-world image classification benchmark consisting of cropped digit images collected from Google Street View. It contains over 600,000 RGB images of size $32 \times 32$, covering digits from 0 to 9 under diverse lighting conditions, backgrounds, and viewpoints. Compared to synthetic or clean datasets such as MNIST, SVHN exhibits significantly higher visual complexity and noise, making it a more challenging benchmark for evaluating model robustness and representation learning ability. In this work, we follow the standard SVHN classification protocol and use it to assess the effectiveness of temporal modeling in Spiking Trnsformers.

*Table 11.* Performance comparison on the SVHN dataset. All models are trained with batch size 128 for 400 epochs.

| Model | Size | Acc@1 (%) | Heads |
|-------|------|-----------|-------|
| Spikformer (Zhou et al., 2022) | 4-384 | 96.72 | 12 |
| SDT (Yao et al., 2023) | 4-384 | 96.84 | 12 |
| QKFormer (Zhou et al., 2024a) | 4-384 | 96.78 | 12 |
| TIMv1 (Shen et al., 2024) | 4-384 | 96.41 | 12 |
| **TEFormer(ours)** | 4-384 | **96.88** | 12 |

## B.2. Robustness among Size

All size-related experiments were conducted under identical conditions. Owing to the characteristics of the Attention mechanism, the embedding dimension must be an integer multiple of the number of heads. Accordingly, when the embedding dimension is 384, the number of heads is set to 12; when the embedding dimension is 256, the number of heads is set to 8.

*Table 12.* Comparison on CIFAR10 of TEFormer with two baselines under different model sizes.

| Model | Size | Acc@1 | Heads |
|-------|------|-------|-------|
| QKFormer (Zhou et al., 2024a) | 2-384 | 93.60 | 12 |
| | 4-256 | 95.04 | 8 |
| | 4-384 | 95.91 | 12 |
| TIM (Shen et al., 2024) | 2-384 | 94.16 | 12 |
| | 4-256 | 93.11 | 8 |
| | 4-384 | 94.20 | 12 |
| **TEFormer(Ours)** | 2-384 | **94.38** | 12 |
| | 4-256 | **95.33** | 8 |
| | 4-384 | **96.24** | 12 |

