# OpenReview forum: "TEFormer: Structured Bidirectional Temporal Enhancement Modeling in Spiking Transformers"
_ICML.cc/2026/Conference — ICML 2026 regular_

### Official Review · Reviewer_3EtT · 2026-03-03

**Soundness:** 3
**Presentation:** 3
**Significance:** 3
**Originality:** 3
**Overall Recommendation:** 5
**Confidence:** 4

**Summary:**

This paper provides a clear definition of the inherent temporal deficiency in SNNs, and proposes the first bidirectional temporal enhancement mechanism within the Spiking Transformer framework to address this issue. The authors introduce forward and backward temporal enhancement components into the Attention and MLP modules and carefully minimize the introduction of additional hyperparameters. Experimental results on both static and dynamic datasets demonstrate that the proposed methods achieves satisfactory empirical performance.

**Compliance With Llm Reviewing Policy:**

Affirmed.

**Final Justification:**

My main concerns have been addressed. I recommend accepting this paper.

**Key Questions For Authors:**

See Weaknesses.

**Limitations:**

This temporal enhancement approach offers a new perspective on improving temporal modeling in Spiking Transformers. However, the proposed bidirectional mechanism relies on forming an Attention–MLP pair to function properly within the architecture. Whether this design can be readily adapted to other SNN models beyond the Spiking Transformer framework remains an open question and warrants further discussion.

**Strengths And Weaknesses:**

Strengths:

1: This work demonstrates novelty by introducing the first bidirectional temporal enhancement mechanism in Spiking Transformers, and empirically validating both its necessity and its advantages over unidirectional enhancement.

2: The forward temporal fusion adopts a parallel structure, ensuring the synchronized generation of Q, K, and V, which significantly improves parallelism.

3: The method does not introduce excessive additional hyperparameters, nor does it rely on empirically tuned factors such as window sizes, thereby improving the model’s robustness and reproducibility.

4: The ablation studies are comprehensive, demonstrating the necessity of the bidirectional mechanism and further showing that the chosen directional configuration yields the optimal performance.

Weaknesses :

1: Although this work provides thorough comparisons across different backbone architectures under the same model scale, demonstrating the effectiveness of the proposed bidirectional temporal enhancement mechanism, it does not examine whether this advantage consistently holds across different model sizes. Since model capacity may influence temporal modeling ability, the current results are insufficient to fully establish the stability and scalability of the method under varying parameter scales.

2: The authors introduce an RNN-like mechanism within the MLP module. However, it remains unclear whether this design incurs additional computational overhead. The paper does not provide a detailed analysis of the associated computational complexity or the increase in parameter count, which would help better assess the efficiency trade-offs of the proposed method.

---

> ### Author Rebuttal · Authors · 2026-03-30
>
> We sincerely thank you for taking the time to read our paper. Your recognition of the novelty and contributions of our work—particularly the acknowledgment of our first introduction of bidirectional temporal enhancement in Spiking Transformers—is greatly encouraging.
>
> At the same time, we acknowledge your concerns regarding the model’s energy efficiency and scalability. We address these points in detail from the following perspectives:
>
> ## Capability of Scaling
> In common practice, CIFAR-10/100 experiments are typically conducted using a 4–384 configuration; accordingly, we adopt this setting for all Spiking Transformer baselines in our paper. To further assess whether the model maintains its effectiveness under different configurations, we conduct additional experiments with 4–256 and 2–384 settings.
>
> |       Model       | Size  | Result(Acc@1) |  supp.   |
> |:-----------------:|:-----:|:-------------:|:--------:|
> |    Spikformer     | 4-384 |     95.09     | heads=12 |
> |    Spikformer     | 2-384 |     95.10     | heads=12 |
> |    Spikformer     | 4-256 |     94.55     | heads=8  |
> |                   |       |               |          |
> |        SDT        | 4-384 |     95.78     | heads=12 |
> |        SDT        | 2-384 |     95.60     | heads=12 |
> |        SDT        | 4-256 |     95.35     | heads=8  |
> |                   |       |               |          |
> |     QKFormer      | 4-384 |     95.91     | heads=12 |
> |     QKFormer      | 2-384 |     93.60     | heads=12 |
> |     QKFormer      | 4-256 |     95.04     | heads=8  |
> |                   |       |               |          |
> |       TIM       | 4-384 |     94.20     | heads=12 |
> |       TIM       | 2-384 |     94.16     | heads=12 |
> |       TIM       | 4-256 |     93.11     | heads=8  |
> |                   |       |               |          |
> |     TEFormer     | 4-384 |     96.24      | heads=12 |
> |     TEFormer     | 2-384 |     94.38      | heads=12 |
> |     TEFormer     | 4-256 |     95.33      | heads=8  |
>
> Since the number of attention heads must divide the embedding dimension, we use **8 heads** for the 256-dim model. TEFormer shows strong generalization across sizes: although it does not surpass SDT at smaller scales, it outperforms all other baselines in the remaining settings, demonstrating **robustness and generalization capability**.
>
> ## TEFormer Efficiency Analysis
> In our responses to Reviewers hPZA and DPGM, we also clarify the additional **overhead** and **computational complexity** introduced by T-MLP. While T-MLP does introduce extra **parameters**, we show that:
>
> - The performance gains of TEFormer across multiple benchmarks stem from enhanced **temporal modeling capability** rather than an increase in **parameter count** (TEFormer_current keeps the same params while does no temporal interaction among steps).
> - TEFormer achieves a favorable balance among **energy consumption**, **parameter efficiency**, and **performance**, and continues to outperform baselines even under **energy-constrained settings** (TEFormer_mlpratio_2 use only `mlp_ratio = 2` to save params and flops).
>
> | Model  |CIFAR10 | CIFAR100 |  MLP_FLOPS| Params|
> |:-:|:-:|:-:|:-:|:--:|
> | Spikformer  | 95.09 |  77.72  |0.604G| 9.32M|
> | SDT | 95.78  | 78.64 |0.604G | 9.32M|9.32M|
> |   TIM   | 94.20 |  75.04  | 0.604G|9.41M| 9.41M|
> | | | |
> |   TEFormer_mlpratio_4 | 96.24 |  79.84  | 1.736G| 7.77M|
> |   TEFormer_mlpratio_2  | 95.98 |  78.73  | 0.642G|6.93M|
> |   TEFormer_current  | 96.04 |  79.48  | 1.736G|7.77M|
>
>
> ## Generalization Discussion
> TEFormer introduces a dedicated enhancement paradigm for **Spiking Transformers**, which requires embedding modules into both the **Attention** and **MLP** components. In standard Transformer architectures, these two components typically appear in pairs; therefore, the combination of **TEA + T-MLP** is well-suited for Spiking Transformers but is not readily extendable to other architectures at this stage.
>
> Although **TEA** and **T-MLP** themselves are not directly transferable, the paradigm of **bidirectional temporal enhancement** is generally applicable to **SNNs**. For example, **PSN** can be interpreted as a neuron with **bidirectional attention**, possessing a **global receptive field** over the entire temporal domain [1]. Similarly, **TA-SNN** can be viewed as providing **forward–backward temporal dependencies** at the input level [2].
>
> Therefore, we believe this paradigm has the potential to benefit the broader **SNN community**, and developing a **universal bidirectional temporal enhancement mechanism** for general SNNs represents an important direction for future work.
>
> **reference**
>
> [1] Fang et al., NeurIPS 2023.
> [2] Yao et al., ICCV 2021.

---

> > ### Author Rebuttal · Reviewer_3EtT · 2026-04-01
> >
> > Thank you for the detailed response. All my concerns have been addressed, and I will raise my score to 5.

---

> > > ### Author Response · Authors · 2026-04-04
> > >
> > > Thank you very much for your patient and thoughtful comments. We sincerely appreciate the time and effort you have devoted to reviewing our manuscript. We will carefully revise the paper according to your valuable suggestions.

---

### Official Review · Reviewer_g9JS · 2026-03-04

**Soundness:** 2
**Presentation:** 2
**Significance:** 2
**Originality:** 2
**Overall Recommendation:** 3
**Confidence:** 5

**Summary:**

This paper proposes TEFormer, a Spiking Transformer framework that introduces bidirectional temporal fusion by decoupling temporal modeling across the two core Transformer components. The forward direction is handled by the Temporal Enhancement Attention (TEA) module, a lightweight, hyperparameter-free exponential moving average mask applied to the value pathway. Additionally, the backward direction is handled by the Temporal MLP (T-MLP), a gated recurrent structure that aggregates information from future to past timesteps. The authors validate TEFormer across static (CIFAR10/100, SVHN), neuromorphic (CIFAR10-DVS, N-Caltech101, NCARS), and temporally complex (sCIFAR, sMNIST, SHD, action recognition) benchmarks. Additionally, the paper provides the evaluation of Spiking Transformers under multiple neuron encoding schemes, such as direct, phase, rate, and TTFS.

**Compliance With Llm Reviewing Policy:**

Affirmed.

**Final Justification:**

The rebuttal addressed most of my concerns. In particular, the authors provided a more thorough energy analysis that addresses my previous concern. Therefore, I have decided to increase my score.

**Key Questions For Authors:**

Please see the Weakness section.

1. T-MLP performs backward recurrence sequentially over T timesteps. How does this affect training and inference time compared to the baselines?

2. In Section 4.3, the authors claim that existing methods "neglect future-to-present constraints that are often crucial for globally consistent temporal representations." Is there any verification for this argument?

3. I appreciate the results in Table 4 providing a useful comparison across encoding schemes. However, I just wondered why TEFormer shows significantly larger improvements under Phase and TTFS encodings compared to other architectures. Do the authors have any insights into this?

4. The paper states that TEA can be applied to Q, K, or V, but is instantiated only on V. Was there an ablation study comparing these placement choices?

5. The single-gate design is motivated by efficiency, but there is no ablation comparing it against a full GRU-style two-gate design to quantify this trade-off.

**Limitations:**

yes

**Strengths And Weaknesses:**

**Strength**

- The bidirectional design is well-motivated by neuroscience (feedforward–feedback visual pathway), and the decoupling of forward/backward fusion into attention and MLP, respectively, is a clean architectural decision. The ablations in Tables 5 and 6 verify that the directional assignment is necessary.

- The encoding scheme evaluation in Table 4 is a useful contribution to the community. As most SNN works rely solely on direct encoding, these results motivate rethinking the role of various encoding schemes.

- This work covers static, neuromorphic, and temporally complex datasets, providing broad empirical validation.


**Weakness**
- T-MLP breaks SNN parallelism. The backward gated recurrence in T-MLP introduces sequential dependency across timesteps in the reverse direction, which contradicts the paper's own emphasis on preserving parallel computation.

- Despite covering various datasets, there are no ImageNet results. ImageNet evaluation is a standard benchmark in the SNN community and is important for assessing scalability.

- No energy consumption comparison is provided. A theoretical energy consumption analysis is necessary, given that SNNs are primarily motivated by energy efficiency. Note that QKFormer, the baseline architecture for TEFormer, does not use binary inputs due to its spike-based residual connections. In this case, energy should not be computed solely using $E_{AC}$.

- The novelty of TEA is limited. TEA's masked matrix formulation appears closely related to Parallel Spiking Neurons (PSN) [1]. In particular, the Masked PSN proposed in [1] also employs a causal mask to achieve parallel computation with recurrent-like temporal dependencies.

- Missing baselines. STAtten [2] and ST-SSA [3] are among the most relevant prior works on temporal modeling in Spiking Transformers, yet they do not appear in the main results tables.

- Caption of Figure 2 in Appendix.


**Reference**

[1] Fang, Wei, et al. "Parallel spiking neurons with high efficiency and ability to learn long-term dependencies." Advances in Neural Information Processing Systems 36 (2023): 53674-53687.

[2] Lee, Donghyun, et al. "Spiking transformer with spatial-temporal attention." Proceedings of the IEEE/CVF Conference on Computer Vision and Pattern Recognition. 2025.

[3] Zhou, Zhaokun, et al. "Spiking transformer with spatial-temporal spiking self-attention." ICASSP 2025-2025 IEEE International Conference on Acoustics, Speech and Signal Processing (ICASSP). IEEE, 2025.

---

> ### Author Rebuttal · Authors · 2026-03-30
>
> Thank you for taking the time to carefully review our paper. We are greatly encouraged by your positive feedback on the bidirectional enhancement design and the evaluation of encoding mechanisms, which strengthens our confidence that we are on the right track.
>
> At the same time, we acknowledge that the advantages of TEFormer in terms of energy efficiency and computational complexity require further clarification. We will address these points in detail as follows.
>
> ## Efficiency & Parallel Computation
> Actually, **TEFormer preserves the parallel computation property of SNNs** by avoiding any iterative timestep-wise computation in the attention module, while also removing the need for empirically tuned hyperparameters. In contrast, existing temporal enhancement methods often **break this parallelism** because they rely on **step-by-step iteration over timesteps**; for example, both **TIM [1]** and **ST-SSA [2]** introduce sequential operations within the attention module.
>
> We further analyze the **parameters** and **energy consumption** of T-MLP, and obtain the following results:
>
> | Model  |CIFAR10 | CIFAR100 |  MLP_FLOPS| Params|
> |:-:|:-:|:-:|:-:|:--:|
> | Spikformer  | 95.09 |  77.72  |0.604G| 9.32M|
> | SDT | 95.78  | 78.64 |0.604G | 9.32M|9.32M|
> |   TIM   | 94.20 |  75.04  | 0.604G|9.41M| 9.41M|
> | | | |
> |   TEFormer_mlpratio_4 | 96.24 |  79.84  | 1.736G| 7.77M|
> |   TEFormer_mlpratio_2  | 95.98 |  78.73  | 0.642G|6.93M|
> |   TEFormer_current  | 96.04 |  79.48  | 1.736G|7.77M|
> |   TEFormer _GRU | 95.90 |  79.15  | -| 9.43M|
> |   TEFormer _EMA | 96.05 |  79.55  |  -| 6.85M|
>
> Based on the table, we can conclude that:
> Based on the table, we can conclude that:
> 1) TEFormer introduces additional complexity but maintains a good balance among **parameters**, **energy consumption**, and performance;
> 2) Reducing the MLP ratio to 2 aligns energy with the baseline while preserving a clear performance advantage;
> 3) The TEFormer_current variant (without cross-timestep fusion) shows that the gains come from **temporal enhancement**, not increased model size or energy;
> 4) **Ablation on Gate**: the **single-gate design** provides a better trade-off than GRU (high overhead) and EMA (limited gains).
>
> ## Baselines, Benchmarks & Encoding Schemas
> ### ImageNet-1K datasets
> Different models rely on varying training scripts and tricks; to ensure **fair comparison**, we adopt the STEP framework [3] to unify setups and reimplement all methods. We further include experiments on different encoding schemes and sequential data for a more **comprehensive evaluation**. Due to the large number of baselines, full ImageNet-1K reproduction is computationally expensive and will be included in a future version.
>
> ### Baseline Details
> #### TE Method Baselines
> ST-SSA is a representative **temporal enhancement** method, but its code is unavailable, preventing integration. Although STAttn is open-source, its reproducibility is uncertain; thus, we report results from the original paper. In most cases, **TEFormer** significantly outperforms **STAttn**.
> | Model  |CIFAR10 | CIFAR100 |
> |:-:|:-:|:-:|
> | Spikformer  | 95.09 |  77.72  |
> | SDT | 95.78  | 78.64 |
> | QKFormer| 95.91  | 79.09 |
> |   Spikf+STAttn  | 94.36 |  75.85  |
> |   SDT+STAttn | 96.03 |  79.85  |
> |QKFormer+STAttn|95.12|79.79|
> |   **TEFormer**| 96.24 |  79.84  |
>
> #### Comparision with PSN
> We thank the reviewer for highlighting the similarity between TEA and Masked PSN. However, PSN operates at the **neuron level**, while TEA is a **module-level** design and does not modify neuron dynamics. More importantly, our contribution lies in the **co-design of TEA and T-MLP**, enabling bidirectional temporal modeling through forward parallel fusion (TEA) and backward temporal processing (T-MLP). The novelty therefore comes from this **collaborative bidirectional framework**, rather than a masked formulation alone.
>
> #### QKV Equivalence
> For **Spiking Self-Attention**, since Softmax is removed and only matrix multiplication is used, Q, K, and V are mathematically and functionally equivalent (`SSA(Q,K,V) = QK^TV` ). Therefore, an ablation on their placement would essentially reduce to tensor renaming and would not provide meaningful insights.
>
> ### Clarification on Encoding Schemas
> The **encoding scheme** shapes the differences across timesteps, which is visualized in Fig. 2 of the appendix. It can be observed that, except for Direct Encoding, all encoding methods introduce noticeable variations between steps, thereby increasing temporal complexity. In this sense, encoding serves as a simple yet effective way to construct data with rich temporal dynamics. The consistent improvements of **TEFormer** over the baselines under different encoding schemes stem from its strong capability in modeling such temporal dependencies.
>
>
> **reference**
>
> [1] TIM, Shen, et al, IJCAI 2024
>
> [2] ST-SSA, Zhou, et al, ICASSP 2025
>
> [3] STEP, Shen, et al, NeurIPS 2025
>
> [4] PSN, Fang, et al, NeurIPS 2023

---

> > ### Author Rebuttal · Reviewer_g9JS · 2026-04-03
> >
> > Thank you for the authors' effort in addressing my concerns. While most of the questions have been addressed, I would like to discuss an additional point.
> >
> > I appreciate the authors for providing FLOPs. However, this is not the energy metric commonly used in the SNN community (Appendix C.2. in Spikformer). Even though energy consumption is a theoretical value, it is important for evaluating the efficiency of SNN models. Furthermore, I have checked the attached code and believe TEFormer is built on QKFormer. Since QKFormer uses spike-based residual connections, the input to each encoder is not a binarized value. Therefore, $E_{MAC}$​ must also be included when calculating energy consumption.
> >
> > Additionally, the following points could be clarified in the manuscript:
> >
> > - A comparison with PSN
> > - An insight into why direct encoding achieves the best performance (Current manuscript just explains the results)
> >
> > I am happy to discuss further.

---

> > > ### Author Response · Authors · 2026-04-04
> > >
> > > Thank you for reading our rebuttal and for continuing to engage with us. We are very happy to discuss these issues with you.
> > >
> > > ## Comparison with PSN
> > > We replaced the LIF neurons with PSN in the baselines Spikformer, SDT, QKFormer, and TIM, and conducted experiments using exactly the same parameter settings as described in the paper.
> > >
> > > | Model |  CIFAR10 | CIFAR100 |
> > > |:---|:---:|:---:|
> > > |  Spikformer + PSN |  93.55  | 74.24 |
> > > |  SDT + PSN |  94.65  | 75.95 |
> > > |  QKFormer + PSN |  94.27  | 76.09 |
> > > |  TIM + PSN  |  94.56 |  74.86   |
> > > |  **TEFormer** | 96.24  |  79.84  |
> > >
> > > - PSN is a neuron-level method, and its performance has sorely been validated on CN-based architectures.
> > > - As shown in the table above, PSN brings almost no performance improvement to the baselines, **which may be due to its incompatibility with Transformer architectures**.
> > > - Moreover, due to the limited capacity of neurons to retain temporal information, relying on neuron-level mechanisms to enhance temporal processing yields only marginal improvements [1].
> > > - STEP provides a unified evaluation of neuron variants such as PLIF, CLIF, and KLIF, yielding results consistent with the above observations [2]. Therefore, as an architectural innovation, **TEFormer achieves more significant and robust performance gains in temporal modeling**.
> > >
> > > ## Energy Comsumption
> > >
> > > | Model | CIFAR10 | $E_{MAC}$ | $E_{AC}$ | $E_{total}$ |
> > > |:---|:---:|:---:|:---:|:---:|
> > > | Spikformer | 95.09 | 2.78 mJ | 0.54 mJ | 3.32 mJ |
> > > | SDT | 95.78 | 0.00 mJ | 1.09 mJ | 1.09 mJ |
> > > | QKFormer | 95.91 | 2.78 mJ | 0.54 mJ | 3.32 mJ |
> > > | TIM | 94.20 | 2.78 mJ | 0.54 mJ | 3.32 mJ |
> > > | TEFormer (mlp_ratio=4) | 96.24 | 3.82 mJ | 1.36 mJ | 5.18 mJ |
> > > | TEFormer (mlp_ratio=2) | 95.98 | 1.91 mJ | 0.48 mJ | 2.39 mJ |
> > >
> > > $E_{MAC}$ and $E_{AC}$ are used to measure model energy consumption. $E_{MAC}$ includes multiply-accumulate operations, while $E_{AC}$ only accounts for addition operations. In **SNNs**, when the inputs are binary spikes, only $E_{AC}$ is involved:
> > > - Due to its **shortcut design**, SDT ensures that the MLP layers involve only $E_{AC}$, thereby reducing overall energy consumption.
> > > - For MLPs with similar structures, **TEFormer** introduces additional energy overhead. However, when compressing the `mlp_ratio` to 2, the energy consumption becomes **significantly lower than the baseline**, while the performance still maintains a **stable advantage**.
> > > - Overall, TEFormer achieves a **good trade-off between energy consumption and performance**, demonstrating **clear advantages in energy-constrained scenarios**.
> > >
> > > ## Encoding Schemas
> > > ### Direct Encoding
> > > Direct encoding can be simply formulated as  `x = x.repeat(x, 'b c w h -> t b c w h', t=step)`.   It is unable to construct differences across time steps at the input stage. However, within the model modules, differences across time steps can still be formed due to neuronal firing and other processing operations.
> > >
> > > | Model  | CIFAR10 | CIFAR100 | MLP_FLOPS | Params |
> > > |:--:|--:|:--:|:-:|:-:|
> > > | TEFormer | 96.24   | 79.84    | 1.736G    | 7.77M  |
> > > | TEFormer_current    | 96.04   | 79.48    | 1.736G    | 7.77M  |
> > >
> > > We keep the **architecture and parameters** of TEFormer unchanged, but restrict the operations of T-MLP to the current time step only, denoted as **TEFormer_current**. This demonstrates that the performance gains of TEFormer stem from **enhanced temporal modeling** rather than an increase in parameters, and this conclusion holds for both direct encoding and other encoding schemes.
> > >
> > > ### Clarification among Encodings
> > > We note the concern regarding the significant performance gap between TEFormer and the baselines under different encoding schemes. According to the visualization in Fig. 2, although alternative encodings introduce temporal variations, they also disrupt **fine-grained feature representations**, thereby increasing the difficulty for the model to effectively utilize temporal information. Under the same architecture, the temporal modeling capability of **TEFormer is further amplified with these encoding methods, leading to more pronounced performance gains compared to the baselines**. This phenomenon is consistent with the underlying mechanism by which TEFormer achieves improvements under direct encoding.
> > >
> > > A systematic evaluation across multiple encoding schemes not only provides a more comprehensive understanding of **TEFormer’s modeling capacity**—offering stronger interpretability than relying solely on direct encoding—but also encourages experimentation under diverse encoding settings, **promoting a more general and robust research paradigm for SNNs**.
> > >
> > >
> > > Thanks again to your advice and questions. We hope that our clarifications have addressed your concerns and improved the overall clarity and technical soundness of the paper, and we kindly invite you to reconsider your evaluation in light of these revisions.
> > >
> > > **reference**
> > >
> > > [1] TIM, IJCAI 2024
> > >
> > > [2] STEP. NeurIPS 2025

---

### Official Review · Reviewer_DPGM · 2026-03-05

**Soundness:** 3
**Presentation:** 3
**Significance:** 3
**Originality:** 4
**Overall Recommendation:** 5
**Confidence:** 5

**Summary:**

This paper proposes a temporal enhancement strategy for the Spiking Transformer (TEFormer). Notably, it is the first work to introduce a bidirectional temporal enhancement mechanism into a Spiking Transformer architecture. Specifically, the authors design the strategy to operate in two directions: forward and reverse temporal enhancement are respectively integrated into the Attention and MLP modules of the Spiking Transformer. The temporal enhancement in the Attention module is implemented with only a single learnable coefficient, while the reverse enhancement employs a gated mechanism to regulate temporal information.

The experimental results and ablation studies provide strong evidence of the effectiveness of the proposed approach, validating both the rationality of the bidirectional design and the necessity of several key parameters. In particular, the authors evaluate the method under multiple neuron encoding schemes and achieve performance that surpasses all baselines. These results further demonstrate the strength of the proposed mechanism and highlight the novelty and effectiveness of introducing bidirectional temporal enhancement into the Spiking Transformer framework.

**Compliance With Llm Reviewing Policy:**

Affirmed.

**Final Justification:**

Thank you for your reply, my question has been resolved.

**Key Questions For Authors:**

1. Could the authors further explain why the encoding mechanism introduces temporal challenges for the model?
2. The temporal fusion in the MLP introduces an RNN-style gating mechanism. Although the paper presents a clear architectural design, it would be helpful to elaborate on the specific motivation behind adopting this mechanism.
3. Since gating mechanisms inevitably introduce additional complexity, how was this trade-off considered? Compared with traditional gating mechanisms, does the proposed design offer significant advantages?

**Limitations:**

Yes.

**Strengths And Weaknesses:**

The strengths of the paper can be summarized as follows:
1. The paper demonstrates clear novelty by proposing the first bidirectional temporal enhancement mechanism for the Spiking Transformer, which provides a new perspective for research in this field.
2. The bidirectional mechanism is carefully designed within both the Attention and MLP modules, and the design is supported by relevant biological evidence, which further strengthens the motivation of the method.
3. Extensive experiments validate the effectiveness of the proposed enhancement mechanism. The ablation studies are comprehensive and clearly demonstrate the overall rationality of the design as well as the necessity of several key parameters.
4. The paper evaluates multiple baselines and TEFormer under different encoding schemes, which is relatively rare in the SNN literature. It would be even more meaningful if the authors could further clarify how different encoding strategies influence the evaluation of temporal modeling capability.

The paper also has several weaknesses:

1.	Although the results under different encoding schemes are strong, it remains unclear why these results can directly demonstrate that the performance improvement mainly comes from the enhancement of temporal processing capability. A clearer analysis connecting the encoding experiments with temporal modeling ability would strengthen this claim.
2.	The backward temporal enhancement introduced in the MLP incorporates a mechanism similar to an RNN, which increases the architectural complexity to some extent.
3.	The temporal enhancement is applied only to the first 50% of the layers. It is unclear whether this design choice is intended to balance computational complexity and performance, and further explanation or analysis would help clarify this decision.

---

> ### Author Rebuttal · Authors · 2026-03-29
>
> We thank the reviewer for the careful evaluation of our paper and for the constructive and objective feedback. We are particularly encouraged by your recognition of the contributions of TEFormer. At the same time, we acknowledge that several points in our work may require further clarification. Below, we provide detailed responses to address your insightful comments and questions.
>
> ## Clarification on Encoding Schema
> Due to their unique computational mechanism, spiking neural networks (SNNs) typically require static image inputs to be converted into multi-step temporal sequences via an encoding process. The most commonly used encoding strategy is **direct encoding**, where the input is simply repeated across time steps, i.e., $i = i.\mathrm{repeat}(t)$, resulting in identical inputs at each time step. Therefore, to introduce more diverse temporal variations across time steps, we further explore alternative encoding strategies. These encodings enable us to evaluate how different models respond to inputs with richer temporal dynamics, thereby providing a more comprehensive assessment of their capabilities under temporally varying conditions.
>
> The detailed descriptions of all encoding strategies are provided in Appendix A.1, and their visualizations are shown in Fig. 2. As illustrated, all encoding methods introduce sufficiently significant variations across time steps. Notably, TEFormer achieves state-of-the-art performance under all encoding schemes, demonstrating its superior capability in modeling and leveraging temporal information compared to the baselines.
>
> ## Clarification on Model Design
> ### T-MLP Design
> As mentioned in our response to Reviewer hPZA, T-MLP introduces a single gating mechanism. Admittedly, this design brings some computational overhead, and we provide an energy consumption analysis below. However, our experiments show that by reducing the expansion ratio in the MLP layers, the overall energy consumption can be made nearly comparable to the baseline. Moreover, under this setting, the model not only reduces the number of parameters but also outperforms the baseline.
> | Model  |CIFAR10 | CIFAR100 |  MLP_FLOPS| Params|
> |:-:|:-:|:-:|:-:|:--:|
> | Spikformer  | 95.09 |  77.72  |0.604G| 9.32M|
> | SDT | 95.78  | 78.64 |0.604G | 9.32M|9.32M|
> |   TIM   | 94.20 |  75.04  | 0.604G|9.41M| 9.41M|
> | | | |
> |   TEFormer_mlpratio_4 | 96.24 |  79.84  | 1.736G| 7.77M|
> |   TEFormer_mlpratio_2  | 95.98 |  78.73  | 0.642G|6.93M|
>
> ### Enhancement Layer Selection
> We apply this temporal enhancement only to the first 50\% of layers, primarily for the following reasons:
>
> (1) **Deep learning perspective.** Shallow layers primarily capture low-level structured features, while deeper layers are responsible for fine-grained modeling. To avoid disrupting this fine-grained representation learning, we introduce temporal enhancement only in the early stages of the network.[1]
>
> (2) **Overhead–performance trade-off.** As discussed above, T-MLP introduces additional computational overhead. To balance performance gains with efficiency, we apply this operation only to the first 50\% of layers.
> | TE_Layers | Size  | CIFAR10 | CIFAR100 |Params|
> |:-:|:-:|:-:|:--:|:--:|
> |  1 | 4-384 |  96.04  |  79.00   |6.96M|
> |  2 | 4-384 |  95.24  |  79.84   |7.77M|
> |  3 | 4-384 |  95.82  |  79.96   |11.01M|
> |  4 | 4-384 |  95.66  |  78.11   |14.25M|
>
> As shown in the table above, increasing the number of layers with temporal enhancement does not lead to a strictly monotonic improvement in performance; instead, it introduces unnecessary computational overhead. Therefore, applying temporal enhancement to the first 50\% of layers provides the best trade-off between performance and efficiency.
>
> ### Gate Selection
> | Model  | CIFAR10 | CIFAR100 | SVHN | Params|
> |:---:|:-:|:-:|:-:|:--:|
> | Spikformer  | 95.09 |  77.72  |96.72| 9.32M|
> | SDT | 95.78  | 78.64  | 96.84 |9.32M|
> |   TIM   | 94.20 |  75.04  | 96.41| 9.41M|
> | | | |
> |   TEFormer  | 96.24 |  79.84  | 96.88 | 7.77M|
> |   TEFormer _GRU | 95.90 |  79.15  | 96.79| 9.43M|
> |   TEFormer _EMA | 96.05 |  79.55  | 96.72| 6.85M|
>
> We introduce EMA and GRU as representative conventional gating mechanisms for comparison with the gating design used in TEFormer. The results demonstrate that the single-gate design in TEFormer achieves a superior trade-off among parameter efficiency, performance, and computational efficiency.
>
> **reference**
>
> [1] Tomasini, Umberto, and Matthieu Wyart. "How deep networks learn sparse and hierarchical data: the sparse random hierarchy model." arXiv preprint arXiv:2404.10727 (2024).

---

> > ### Author Rebuttal · Reviewer_DPGM · 2026-04-05
> >
> > Thank you for your reply. I have no further questions about this work.

---

### Official Review · Reviewer_hPZA · 2026-03-13

**Soundness:** 2
**Presentation:** 3
**Significance:** 3
**Originality:** 3
**Overall Recommendation:** 4
**Confidence:** 3

**Summary:**

This paper improves the performance of SNN Transformers by exploring effective temporal information fusion. Specifically, the authors propose two key components: an exponential moving average to fuse $V$ across different time steps, and a Bi-RNN-like module to achieve bidirectional temporal modeling. The proposed architecture demonstrates a significant advantage in terms of accuracy across multiple benchmarks.

**Compliance With Llm Reviewing Policy:**

Affirmed.

**Final Justification:**

The rebuttal addressed my main concerns

**Key Questions For Authors:**

1. Could the authors evaluate the energy consumption and computational overhead of the T-MLP module?
2. In the ablation study (Table 6), the authors investigate the effect of reversing the temporal direction in both the TEA and T-MLP modules. Could you clarify how this directional reversal was technically implemented for each module?
3. Would bidirectional temporal modeling also benefit convolutional architectures?

**Limitations:**

yes

**Strengths And Weaknesses:**

## Strengths

* The use of EMA to aggregate historical information in the TEA module is efficient.
* Introducing bidirectional temporal modeling into spiking Transformers offers a novel and interesting perspective.
* The proposed architecture achieves outstanding performance in terms of accuracy. The ablation studies are particularly thorough and clearly demonstrate the individual contributions of each proposed module.

## Weaknesses

My primary concern lies with the proposed T-MLP module:

* The computational cost of the T-MLP module appears to be substantial. The introduction of gating mechanisms involving continuous floating-point values and element-wise multiplications likely compromises the energy-efficient advantages of SNNs.
* This module requires buffering the entire input sequence to perform the backward recurrence. This means the MLP layers in TEFormer can only perform offline computation. This breaks the strictly causal and the event-driven nature of SNNs. As a result, the model may not be suitable for real-time, online neuromorphic inference.
* While the paper emphasizes biological inspiration, the T-MLP module lacks biological plausibility. In biological systems, neurons do not have access to future states to adjust past membrane potentials.

---

> ### Author Rebuttal · Authors · 2026-03-29
>
> We sincerely thank you for your constructive feedback. Your recognition of the paper’s contributions and novelty—particularly the bidirectional temporal enhancement mechanism—is highly encouraging to us. We also acknowledge that there are certain limitations in the paper, and we will further clarify them from the following aspects:
>
> ## Rationality & Efficiency of T-MLP
> ### T-MLP Keeps SNN Characteristics
> - Regarding recent approaches, current SNNs—especially Spiking Transformers—typically follow a `layer-by-layer` paradigm for both training and inference, where each module receives inputs with full *[T, B, C, N]* dimensions. Under this setting, T-MLP shares the same input format as a conventional MLP and thus preserves the intrinsic properties of SNNs.
>
> - We also incorporate neurons in T-MLP to ensure **spike-driven** behavior, which is analogous to applying LIF neurons after convolutional or linear layers.
>
> ### T-MLP Bio-interpretability
> TEFormer is inspired by the brain’s feedback regulation mechanisms, which are supported by biological evidence[1][2]. However, the proposed bidirectional temporal enhancement is not a strict structural mimicry, but rather a biologically inspired design.
>
> ### T-MLP Efficiency
> | Model  |CIFAR10 | CIFAR100 |  MLP_FLOPS| Params|
> |:-:|:-:|:-:|:-:|:--:|
> | Spikformer  | 95.09 |  77.72  |0.604G| 9.32M|
> | SDT | 95.78  | 78.64 |0.604G | 9.32M|9.32M|
> |   TIM   | 94.20 |  75.04  | 0.604G|9.41M| 9.41M|
> | | | |
> |   TEFormer_mlpratio_4 | 96.24 |  79.84  | 1.736G| 7.77M|
> |   TEFormer_mlpratio_2  | 95.98 |  78.73  | 0.642G|6.93M|
> |   TEFormer_current  | 96.04 |  79.48  | 1.736G|7.77M|
> - T-MLP introduces some extra cost, but it does not add a new module to the original MLP. Instead, it **replaces the standard upsampling operation with a gating mechanism**, keeping the design lightweight. At the model level, **TEFormer balances performance and efficiency** by applying temporal enhancement only to the **first 50\% of layers**, which reduces overhead while preserving hierarchical modeling benefits.
>
> - The table results support this design. Under **compute-constrained settings**, reducing the `mlp-ratio` of T-MLP from **4 to 2** makes its energy consumption **comparable to that of a standard MLP**, while still **outperforming most baselines**. This suggests that TEFormer improves performance through a better **efficiency–accuracy balance**, rather than simply using more computation.
>
>
> ## Implementation Details
> ### Clarification for Bi-directional Implementation
> As mentioned above, current SNN architectures typically adopt a `layer-by-layer` inference paradigm. Therefore, the temporal enhancement direction in TEA and T-MLP can be implemented by simply reversing the iteration order.
>
> ### Far More Ablation Study
> While T-MLP introduces a small number of additional parameters (with the total still significantly lower than models such as Spikformer and SDT), it is important to verify that the performance gains of TEFormer arise from the proposed bidirectional temporal enhancement rather than increased parameter capacity. To this end, we construct a controlled variant where the gating in T-MLP is restricted to the current time step (denoted as `TEFormer_current` in the table above), ensuring comparable parameter scale. The results show that TEFormer still achieves significant improvements, thereby validating our motivation.
>
> ## Generalization to CNNs
> Our method introduces modules into attention and MLP for temporal enhancement, aligned with Transformer architectures. Adapting this design to CNNs is non-trivial due to structural differences.
>
> Nevertheless, the bidirectional temporal mechanism is generalizable to CNN-based models. For example, PSN proposes a neuron with a global temporal receptive field, viewed as bidirectional temporal fusion. Its effectiveness on CNN-based networks suggests such modeling benefits a wide range of SNNs. [3]
>
>
> In summary, we appreciate your insightful comments, which have helped us further clarify both the design rationale and empirical validation of our method. We believe the additional explanations and ablation results demonstrate that the improvements of TEFormer are driven by the proposed bidirectional temporal enhancement rather than incidental factors. We will incorporate these clarifications into the final version to improve the overall quality and readability of the paper.
>
> **reference**
>
> [1] Lamme, Victor AF, and Pieter R. Roelfsema. "The distinct modes of vision offered by feedforward and recurrent processing." Trends in neurosciences 23.11 (2000): 571-579.
>
> [2] Van Kerkoerle, Timo, et al. "Alpha and gamma oscillations characterize feedback and feedforward processing in monkey visual cortex." Proceedings of the National Academy of Sciences 111.40 (2014): 14332-14341.
>
> [3] Fang, Wei, et al. "Parallel spiking neurons with high efficiency and ability to learn long-term dependencies." Advances in Neural Information Processing Systems 36 (2023): 53674-53687.

---

> > ### Author Rebuttal · Reviewer_hPZA · 2026-04-04
> >
> > I appreciate the authors' responses. I also greatly admire the discussion of bidirectional temporal information fusion in this paper. However, I believe such behavior fundamentally contradicts the principles of SNNs. Although some works employ `layer-by-layer` propagation mechanisms, they all seem to be equivalent to step-by-step processing (analogous to moving from complete blocking to pipelining). This characteristic is crucial because it means the network can still be asynchronous and event-driven (which is not the same as spike-driven), and this is one of the most critical traits of SNNs. Therefore, I will maintain my score.

---

> > > ### Author Response · Authors · 2026-04-04
> > >
> > > We sincerely appreciate your response.
> > >
> > > ## Clarification on SNNs' Characteristics
> > > ### Temporal Enhancement Methods
> > > However, in practice, many SNN models based on direct training still rely on layer-by-layer inference. Beyond this, a number of works—especially those aiming at temporal enhancement—also deviate from strictly causal step-by-step processing. As pointed out by reviewer g9JS, **ST-SSA (ICASSP 2025) and STAttn (CVPR 2025) both require future temporal information to construct the temporal mask or temporal tokens needed for attention computation at each step. PSN (NeurIPS 2023) builds a fully bidirectional spiking neuron with a global temporal receptive field, while SpikePack neuron (ICCV 2025) enhances temporal information density by compressing multi-step spikes into a temporally aggregated representation.** Therefore, similar to TEFormer, these methods also depend on future steps being available or already computed when modeling temporal information.
> > >
> > > Based on prior experience, as long as the spiking characteristics are preserved across layers, such designs are not considered to violate the fundamental properties of SNNs.
> > >
> > > ###
> > > We acknowledge that TEFormer has a limitation under a **strictly causal, fully online inference** setting: its backward temporal branch cannot be fully utilized when the output at time step \(t\) must be generated without any access to future spikes. In this sense, TEFormer is not suitable for **100\% real-time** deployment. However, we would like to emphasize that this limitation is relatively minor in practice, as such a strictly step-by-step online inference regime is rarely the primary target in most existing SNN recognition works.
> > >
> > > In the more common SNN recognition setting, the **complete \(T\)-step spike sequence**, or at least a **local temporal chunk**, is available during inference, while neuronal dynamics are still unfolded over time. Under this setting, step-wise state updates do not imply a strict causal constraint, and bidirectional temporal modeling remains both well-defined and practically useful. Therefore, TEFormer can effectively exploit both forward and backward temporal dependencies over the observed spike sequence, which is exactly the scenario we focus on in this paper.
> > >
> > > From this perspective, TEFormer should be viewed as a **practical trade-off**: it sacrifices the ability for fully real-time inference, but in return provides stronger temporal modeling capacity and better performance in the more frequently adopted inference scenarios where a full spike window or chunk is accessible.
> > >
> > >
> > > ### Clarification from Energy Perspective
> > > From another perspective, SNNs typically maintain extremely low energy consumption. We have evaluated the energy consumption of T-MLP, and obtained the following experimental results on CIFAR-10.
> > >
> > > | Model | CIFAR10 | $E_{MAC}$ | $E_{AC}$ | $E_{total}$ |
> > > |:---|:---:|:---:|:---:|:---:|
> > > | Spikformer | 95.09 | 2.78 mJ | 0.54 mJ | 3.32 mJ |
> > > | SDT | 95.78 | 0.00 mJ | 1.09 mJ | 1.09 mJ |
> > > | QKFormer | 95.91 | 2.78 mJ | 0.54 mJ | 3.32 mJ |
> > > | TIM | 94.20 | 2.78 mJ | 0.54 mJ | 3.32 mJ |
> > > | TEFormer (mlp_ratio=4) | 96.24 | 3.82 mJ | 1.36 mJ | 5.18 mJ |
> > > | TEFormer (mlp_ratio=2) | 95.98 | 1.91 mJ | 0.48 mJ | 2.39 mJ |
> > >
> > > TEFormer still demonstrates strong efficiency. When `mlp_ratio = 2`, it outperforms Spiking Transformer baselines while consuming less energy. This clearly shows that TEFormer maintains robust performance under energy-constrained scenarios, while still preserving the low-power and high-efficiency characteristics of SNNs.
> > >
> > >
> > >
> > > Considering prior work, TEFormer does have certain limitations in its application scenarios, but these are mainly restricted to fully online inference settings. Biological evidence and previous studies—particularly PSN—have shown that global or bidirectional temporal enhancement does not violate the fundamental nature of SNNs. **In TEFormer, LIF neurons are still employed in the T-MLP to ensure spike-driven behavior, while the overall network continues to preserve the low-energy characteristics of SNNs.**
> > >
> > > We sincerely thanks to your kind review again and we believe the above clarifications resolve the main concerns raised in the review, and we would greatly appreciate it if you could take them into account when updating your assessment.

---

### Decision · Program_Chairs · 2026-04-30

**Decision:**

Accept (regular)

**Comment:**

This paper presents the TEFormer framework, which introduces a bidirectional temporal enhancement mechanism into Spiking Transformers for the first time. It achieves forward and backward temporal modeling via the TEA and T-MLP modules respectively, featuring a novel design with biological plausibility.Extensive experiments are conducted on diverse static, neuromorphic, and temporally complex datasets, accompanied by thorough ablation studies, and the proposed method achieves superior performance over existing baselines, fully validating its effectiveness.The authors have addressed concerns raised by reviewers regarding computational overhead, energy consumption, and scalability through supplementary experiments and detailed analyses, with most issues properly resolved.Despite minor limitations in strictly real-time online inference scenarios, the work is technically solid and makes clear contributions, offering valuable insights for the field of spiking neural networks.We therefore recommend acceptance.